# SGLT2 Inhibitors and the Risk of Contrast-Associated Nephropathy Following Angiographic Intervention: Contradictory Concepts and Clinical Outcomes

**DOI:** 10.3390/ijms251910759

**Published:** 2024-10-06

**Authors:** Samuel N. Heyman, Doron Aronson, Zaid Abassi

**Affiliations:** 1Department of Medicine, Hadassah Hebrew University Hospital, Mt. Scopus, Jerusalem 91240, Israel; 2Department of Cardiology, Rambam Health Care Campus, Haifa 3109601, Israel; d_aronson@rambam.health.gov.il; 3Department of Physiology, Bruce Rappaport School of Medicine, Technion, Haifa 3525433, Israel; 4Department of Laboratory Medicine, Rambam Health Care Campus, Haifa 3109601, Israel

**Keywords:** SGLT2i, coronary interventions, radiocontrast nephropathy, chronic kidney disease, hypoxia

## Abstract

The use of SGLT2 inhibitors (SGLT2is) has been found in large clinical studies to slow the progression of chronic kidney disease (CKD) and to lower the risk of acute kidney injury (AKI). Recent reports suggest that SGLT2is may also reduce the likelihood of developing radiocontrast-associated nephropathy (CAN) following contrast-enhanced imaging and intravascular interventions. This review underscores potential pitfalls and confounders in these studies and calls for caution in adopting their conclusions regarding the safety and renoprotective potency of SGLT2is, in particular in patients at high risk, with advanced CKD and hemodynamic instability undergoing coronary intervention. This caution is particularly warranted since both SGLT2is and contrast media intensify medullary hypoxia in the already hypoxic diabetic kidney and their combination may lead to medullary hypoxic damage, a principal component of CAN. Further studies are needed to evaluate this dispute, particularly in patients at high risk, and to reveal whether SGLT2is indeed provide renal protection or are hazardous during contrast-enhanced imaging and vascular interventions.

## 1. Introduction

The introduction of SGLT2 inhibitors (SGLT2is) has revolutionized the management of type II diabetes mellitus (NIDDM), achieving, in addition to an outstanding attenuation of all metabolic derangements, remarkable cardiovascular and renal protection. SGLT2is were found to retard the progression of chronic renal failure (CKD) among patients with and without diabetes and to reduce the incidence of acute kidney injury (AKI) [1,2,3]. The outstanding cardiovascular advantages of SGLT2is have recently been reported, even immediately following coronary interventions for acute myocardial ischemia [4], and more and more of these patients are currently discharged prescribed with SGLT2is.

Natriuresis, coupled with reduced trans-glomerular pressure and improved renal cortical oxygenation, is a principal mechanism likely responsible for acute and chronic renoprotection, along with a better cardiac performance in patients with cardiorenal syndrome. However, SGLT2is intensify renal medullary physiologic hypoxia, generating concern regarding their safety, especially under conditions that predispose to medullary hypoxic injury, such as the exposure to non-steroidal anti-inflammatory drugs (NSAIDs) or iodinated radiocontrast media (CMs) [5,6]. This concern has been supported by sporadic descriptions and by adverse event report systems [7].

From these perspectives, recent clinical reports [8,9,10,11,12,13] and meta-analyses [14] that have unexpectedly indicated renal safety and even advantages in patients given radiocontrast media during coronary interventions while on SGLT2is are highly important and bring about the need for the re-evaluation of basic physiologic concepts.

This review focuses on the impact of diabetes, CMs and SGLT2is on the renal parenchymal oxygenation profile and on the role that medullary hypoxia plays in the pathogenesis of contrast-associated nephropathy (CAN). The contradicting messages indicating renal protection by SGLT2is in patients undergoing coronary interventions will be discussed by a critical review of these studies, underscoring potential pitfalls in the assessment of the true incidence of CAN in these settings. We shall further propose mechanisms that may explain the controversies between physiologically expected and documented renal outcomes in patients undergoing coronary interventions, using contrasting media while on SGLT2is.

## 2. Renal Medullary Hypoxia, Diabetes and the Pathogenesis of CAN

The renal medulla normally functions at a low ambient pO_2_, around 25 mmHg, reflecting both limited regional blood supply and oxygen shunting across the vasa recta and intense oxygen consumption for tubular transport, carried out principally by thick ascending limbs (TALs) and S3 segments in the outer medulla [15]. Oxygen extraction at this region is near maximal, exceeding 79%, forming an outer medullary “anginal syndrome”, where any decline in the vasa recta blood flow or the enhanced transport workload may jeopardize these nephron segments engaged in intense transport activity. Compound physiologic processes maintain an outer medullary oxygenation balance by matching regional blood flow and oxygen consumption. Nitric oxide, prostaglandin, dopamine and adenosine are principal mediators that take part in these processes, governing the glomerular filtration rate, the sodium delivery to distal nephron segments, the tubular ion transport rate and the vasa recta blood flow. Indeed, manipulations that disrupt these systems lead to outer medullary hypoxic injury, particularly noted in TALs, and, to a lesser extent, in S3 segments [16]. By contrast, inhibition of the transport activity, for instance by loop diuretics, improves medullary oxygenation [17] and attenuates outer medullary hypoxic damage [18,19]. In the same fashion, paradoxically, the outer medulla is protected by global renal ischemia, as happens in experimental renal warm ischemia reflow models, as glomerular filtration (hence tubular transport) ceases. This leads to TAL preservation (a nephron segment adjusted to anaerobic glycolysis, resistant to hypoxic injury as long as the transport activity is halted), whereas proximal tubular damage increases (a nephron segment dependent on aerobic glycolysis) [16]. Enhanced tubular transport in remnant functional nephrons, along with disruption and rarefaction of regional microcirculation, is believed to form a feed-forward loop of hypoxia, injury and fibrosis, leading to progressive CKD [20,21].

Diabetes is associated with an intensification of the physiologic medullary hypoxia [22,23,24], related to enhanced oxygen expenditure for tubular transport, and disrupted microcirculation. An excess formation of reactive oxygen species in the diabetic kidney likely further contributes to the generation of mismatched regional oxygen supply and demand. Importantly, even a sub-optimal restoration of euglycemia (mean HbA1C 7.3%) seems to preserve renal parenchymal oxygenation, as illustrated non-invasively by a blood oxygen-level-dependent (BOLD) MRI [25]. However, ongoing pronounced hypoxia likely contributes to CKD progression [21], which conceivably predisposes to AKI in patients with diabetes [26].

CMs markedly intensify medullary hypoxia [27], as shown by oxygen microelectrodes in rodents [28,29] or by BOLD MRI in humans [30]. Intensified outer medullary hypoxia occurs both by the reduction of the vasa recta blood flow and oxygen supply and by a transient increase in the tubular transport workload. The enhanced generation of reactive oxygen species further disrupts the matching of renal oxygen demand and supply and likely intensifies hypoxic injury [31]. Hypoxic tubular damage appears shortly after the administration of CMs in isolated perfused kidneys [32] and in vivo in rats predisposed to medullary hypoxic injury, principally affecting TALs and S3 tubular segments in the outer medulla [28,32,33,34,35]. A gradient pattern of hypoxic injury develops in these models, most pronounced in the mid-inter-bundle zone at the deep (inner stripe) layer of the outer medulla, and away from vascular bundles, where hypoxic stress is maximal. This experimental injury pattern is attenuated by the concomitant inhibition of tubular transport (and oxygen consumption) by TAL, as shown in an animal model of CAN [18]. As summarized in depth [36,37], intensified renal medullary hypoxia, along with the excess formation of oxygen free radicals in the diabetic kidney, likely bring about the particular susceptibility of patients with diabetes to CAN, with an intensification of regional hypoxic and tubulo-toxic injury. In fact, diabetes follows CKD as a principal risk factor for CAN, doubling its incidence at any given level of baseline renal function before coronary intervention [38].

The role of outer medullary hypoxic injury in CAN is further illustrated by a rise in neutrophil gelatinase-associated lipocalin (NGAL), a renal injury biomarker specific to distal tubular injury, including TALs. This is well illustrated in one of the earliest clinical trials determining urine and plasma NGAL, where a very early rise in renal and plasma NGAL predicted evolving CAN in children with congenital heart disease undergoing cardiac imaging with contrast media [39]. Anemia predisposing to CAN [40] and its reduced incidence in patients undergoing coronary interventions with supplemented oxygen [41] may also underline the importance of renal hypoxia in the pathogenesis of CAN.

To conclude, physiologic medullary hypoxia, intensified by SGLT2is, plays a central role in AKI in general, and especially in the diabetic kidney. Medullary hypoxic injury by CMs is considered to be central and upstream to most identified mechanisms believed to take part in CAN [42].

## 3. SGLT2is and Renal Oxygenation: Aggravating Renal Medullary Hypoxia

Glucose tubular transport, coupled with sodium reuptake, takes place in cortical proximal tubular segments by SGLT1 and SGLT2 transporters. SGLT inhibition reduces renal cortical oxygen expenditure for tubular transport not only by blocking sodium/glucose co-transport, but also through the inhibition of e Na1-H1 antiporter (NHE3) [43]. The dual transport inhibition reduces cortical oxygen consumption and improves ambient pO_2_ [5], as shown in rodents following the non-selective inhibition of SGLT [22]. Selective SGLT2is are expected to do the same, as SGLT2 is the principal SGLT isoform. Unexpectedly, however, cortical pO_2_, determined non-invasively by BOLD MRI, remained unchanged in NIDDM patients [44], perhaps because of an up-regulation of the SGLT1 transporter. As opposed to improved or unchanged cortical oxygenation, SGLT inhibition consistently intensifies medullary hypoxia, both in animal studies and in humans [22,44]. This likely occurs since reduced sodium reabsorption in proximal tubules leads to its enhanced delivery to distal nephron segments, resulting in enhanced sodium transport and oxygen consumption principally in TALs [5]. Erythrocytosis, noted with the initiation of SGLT2i treatment, is conceivably triggered by intensified hypoxia at the cortico-medullary junction [45], with the induction of erythropoietin synthesis by peritubular mesenchymal cells, triggered by hypoxia-inducible factor (HIF)-2 [46].

The potential of SGL2is to inflict AKI with selective hypoxic outer medullary injury is illustrated by increasing plasma and urinary NGAL on admission in patients with diabetes hospitalized with AKI, while the corresponding levels of kidney ischemia molecule (KIM)-1 (a marker of proximal tubular injury) remains comparable to those of other patients with diabetes hospitalized without AKI [47].

Concern about the exacerbation of medullary hypoxia by SGLT2is led us to call for caution regarding their use in patients subjected to iatrogenic decline in medullary oxygenation, such as the use of NSAIDs or during radiocontrast studies [6]. Unexpected contradicting outcomes of recently published clinical reports, outlined below, prompted the writing of this review.

## 4. Clinical Studies Indicating Reduced Risk of CAN in Patients on SGLT2is Undergoing Coronary Interventions

Several recently published studies, summarized in Table 1, indicate that patients undergoing coronary interventions while on SGLT2is are not at a greater risk to develop CAN. On the contrary, they all indicate a lower likelihood to develop CAN compared with patients not receiving SGLT2is.

Kültürsay et al. [10] conducted a single-center retrospective analysis of renal outcomes in patients with diabetes with ST-elevation myocardial infarction (STEMI) undergoing percutaneous coronary interventions (PCIs), comparing 130 and 165 SGLT2i-treated and non-treated patients, respectively. They reported a 14% lower incidence of CAN (*p* = 0.028), using doubly robust inverse probability weighted regression with balanced covariates. They excluded patients with advanced renal failure (estimated glomerular filtration rate [eGFR] < 30 mL/min/1/73 m^2^), those with cardiogenic shock or patients managed by insulin. SGLT2i treatment apparently was not interrupted throughout the hospitalization (personal communication).

Santos-Gallego et al. [13] also carried out a retrospective single-center study, analyzing renal outcomes in patients with diabetes who underwent PCI. Fifty-two patients chronically treated with SGLT2is were compared with a matching number of non-users. The two groups were comparable regarding basic clinical and procedural characteristics, including kidney function, hemodynamic status and interventional details. The incidence of CAN was significantly lower in patients on SGLT2is compared with those not treated with SGLT2is (3.8 vs. 17.3%, *p* < 0.05). The average rise in plasma creatinine (pCr) was also lower in the former group (0.11 ± 0.15 vs. 0.29 ± 0.18 mg/dL, *p* < 0.05), with a smaller reduction in eGFR (−4.9 ± 4.1 vs. −9.1 ± 4.5, *p* < 0.05). Noteworthy, there was no reference to the extent of myocardial injury or the type of SGLT2i used, and whether this treatment was interrupted during admission.

Özkan and Gürdoğan [12] conducted a large observational study in patients with diabetes with non-ST elevation-acute coronary syndrome undergoing coronary intervention, looking at the impact of SGLT2is on renal outcomes. Their cohort included 312 patients, of whom 104 were using SGLT2is. Patients with advanced CKD were again excluded, as were patients with heart failure. CAN developed in 30.8% and in 13.5% of patients treated or untreated with SGLT2is, respectively (*p* = 0.03), suggesting substantial nephroprotection provided by SGLT2is. Importantly, treatment with SGLT2is was interrupted throughout hospitalization (personal communication), excluding a confounding impact of increased intraglomerular pressure with a restoration of glomerular hyperfiltration. Multivariate analysis revealed that the use of SGLT2is predicted nephroprotection (OR 0.41, 95% CI 0.142–0.966, *p* = 0.004), while the duration of diabetes and lower eGFR at baseline were the best predictors of CAN.

Paolisso et al. [9,49] examined the impact of SGLT2is (started at least 3 months before hospitalization) on the development of CAN in patients with diabetes with STEMI or non-STEMI undergoing PCI. This retrospective single-center study included 646 patients with diabetes, 111 of them treated with SGLT2is. The patient population was relatively old (mean age 70 years), and among the exclusion criteria were eGFR < 30 mL/min/1.73 m^2^, severe valvular heart disease and severe anemia. No explicit detail was provided whether the SGLT2is were suspended immediately before the procedure and if and when they were re-administered. The use of SGLT2is was associated with a lower occurrence of CAN (5.4% vs. 13.1%), re-hospitalization, cardiovascular death and arrhythmic burden (all *p* < 0.05). These results, again, suggest that the use of SGLT2is by patients with diabetes undergoing PCI for AMI is associated with better cardiovascular and renal outcomes during hospitalization and in the long run.

Comparable results were reported by Bernardini et al. [50], appearing in an abstract form, who conducted a retrospective study of patients undergoing PCI. Included were 136 patients with diabetes treated with SGLT2is, GLP-1 agonists or DPP-4 inhibitors. They were compared with 136 patients with diabetes managed by standard antidiabetic therapy not including these agents, and with an additional group of 136 patients without diabetes. The mean follow-up and exclusion criteria were not reported. The incidence of CAN, determined over 48 h post PCI was 5.1% in the standard antidiabetic therapy group, 3.8% in the group managed by the newer antidiabetic therapies and 2.9% in patients without diabetes. However, the inclusion of various drug classes in the group managed by the new generation of antidiabetic medications precluded the assessment of specific impacts of SGLT2is. For that reason, this work is not included in Table 1.

In an additional retrospective single-center study, Hua et al. [8] explored whether SGLT2i use (dapagliflozin, canagliflozin and empagliflozin) for at least 6 months exerted a nephroprotective effect against CAN in patients with diabetes undergoing elective coronary imaging and interventions. In total, 242 patients on SGLT2is were compared with 242 propensity score matched controls undergoing similar invasive cardiac procedures. Patients with AMI requiring emergency PCI were excluded, as were patients who were hemodynamically unstable, individuals with CKD, heart failure (LVEF < 40%), hepatic dysfunction or poor glycemic control (HbA1c > 8%). Excluded were patients administered with large volumes of contrast media (>400 mL), and subjects with identified other causes of AKI. The incidence of CAN was 63% lower in SGLT2i users [OR: 0.37 (95% CI: 0.18–0.68; *p* = 0.01] before and after adjustment. The authors concluded that SGLT2is are renoprotective in patients with diabetes undergoing elective PCI. However, as outlined below, the weakness of this carefully designed study was the exclusion of the most clinically relevant patients predisposed to CAN, for instance, individuals with advanced CKD and hemodynamic instability. This formed the basis for the unaccepted lack of predictive value of CKD by multivariate regression analysis, the most prominent risk factor for CAN. An additional puzzling outcome was the lack of a class effect, with renal protection apparently noted by dapagliflozin and canagliflozin but not with empagliflozin. Also absent were technical details, such as regarding a possible withholding and resumption of SGLT2is before and after the elective procedure, or the regimens adopted for fluid replacement for the two compared groups.

As opposed to the studies described above, all suggesting renal protection in patients treated with SGLT2is during cardiac interventions, one small, though careful prospective, study reached a different conclusion. Feitosa et al. [11] conducted an open-label randomized (1:1) single-center pilot study with a follow-up of 30 days on patients with diabetes undergoing elective PCI. The clinical outcome of 22 patients given empagliflozin 5 mg/day was compared with that of subjects not treated by SGLT2is. Treatment with empagliflozin was initiated at least 15 days before PCI in the intervention group and was maintained until the end of the follow-up period. Serum NGAL was determined 6 h after PCI and sCr was checked at baseline and at 24 h and 48 h after the procedure. The two groups were comparable regarding the primary outcomes (post PCI NGAL and sCr), with both indices somewhat higher in the SGLT2i group, not suggesting nephroprotection. Noteworthy, patients with a high risk of advanced CKD were, again, excluded.

Meregildo-Rodriguez et al. [14] summarized the above-mentioned studies in a meta-analysis, encompassing 2572 patients with DM undergoing coronary interventions, including 512 patients managed with SGLT2is. Overall, there were 289 patients reported with CAN. The data analysis indicates that SGLT2Is given to patients with diabetes undergoing coronary interventions may reduce the risk of developing CAN by up to 63% (RR 0.37; 95% CI 0.24–0.58). Statistical heterogeneity was not significant (I2 = 0%, *p* = 0.91). The assessment of certainty of the evidence of this systematic review and meta-analysis, according to the GRADE criteria, was found as moderate.

Çabuk and Hazır [48] examined the impact of SGLT2 inhibitors (empagliflozin or dapagliflozin) for at least 6 months on the development of CAN in patients with type II diabetes undergoing CAG or PCI. This cross-sectional and single-center study included 345 patients with type II diabetes, 133 of them treated with SGLT2is. The patient population was of the mean age of 62–64 years, and among the exclusion criteria were eGFR < 30 mL/min/1.73 m^2^, severe anemia and patients who were using SGLT2 inhibitors for less than 6 months. CA-AKI incidence was significantly lower in patients using SGLT2 inhibitors (9.0%) compared with non-users (26.4%, *p* < 0.001). Moreover, the duration of hospitalization was significantly longer in SGLT2i users (3.25 (±2.03) days) than in non-users (2.54 (±1.39) days) (*p*  =  0.001). These results, again, suggest that the use of SGLT2is by patients with diabetes undergoing CAG or PCI is associated with better cardiovascular outcomes and duration of hospitalization.

Basutkar et al. [51] summarized four of the above-mentioned studies in a meta-analysis encompassing a total of 1968 patients. The mean and standard deviation values of the serum creatinine levels at 48 h from only two included studies were considered for analysis. Among a total of 1130 patients, 353 received SGLT2is. The use of SGLT2is conferred substantial beneficial effects on kidney function, as was evident by a significant reduction in levels of SCr 48 h post PCI procedure in the patients who received SGLT2is compared with the patients who were not prescribed SGLT2is, among T2DM with CAD patients (MD −9.57; 95% CI −18.36, −0.78; *p*-value 0.03). Similarly, a substantial reduction in the SCr levels at 72 h was found among the patients who received SGLT2is (MD −14.40; 95% CI −28.57, −0.22; *p*-value 0.05). The data analysis indicates that SGLT2Is given to patients with diabetes undergoing coronary interventions may reduce the risk of developing CAN by up to 63% (RR 0.37; 95% CI 0.24–0.58).

## 5. Contradictory Clinical and Anticipated Outcomes Regarding the Renal Safety of SGLT2is during CM Studies: Plausible Explanations

We have previously called for caution regarding the use of SGLT2is in patients subjected to CMs, since both interventions lead to the intensification of medullary hypoxia, already aggravated in the diabetic kidney [5,6]. Our concern addresses the risk of developing CAN, with the development of hypoxic outer medullary injury. However, recently published clinical studies contradict our concern, most of them indicating preservation rather than deterioration of kidney function following cardiac intravascular intervention/radiocontrast studies. How can one explain this unexpected controversy?

One explanation might be that **confounders may conceal the true incidence of CAN**. We [52,53] and others [54] have noticed that a substantial number of hospitalized patients undergoing contrast studies display improved post-imaging kidney function with declining creatinine over 72 h. In fact, the likelihood of recovering kidney function following contrast-enhanced imaging is roughly three times higher than that of AKI, conceivably reflecting factors unrelated to the exposure to CMs, such as IV fluids, hemodynamic stabilization, successful management of infection and restoration of additional organ dysfunction [53]. These factors, affecting GFR by means unrelated to the radiocontrast studies, might lead to the underestimation of the true incidence of renal parenchymal injury, as we do not routinely determine renal biomarkers or tubular dysfunction [55]. Indeed, when looking at biomarkers of renal tubular injury, for instance NGAL levels, one may detect “subclinical AKI” without changes in plasma creatinine, as reported following coronary interventions [56]. Specifically, patients on SGLT2is might display improved GFR upon transient withholding of these medications, related to the restoration of hyperfiltration [5]. Furthermore, such patients may receive higher volumes of fluids in order to compensate for osmotic diuresis with polyuria.

Patient selection in the studies quoted above may be an additional confounder. Noteworthy, patients with advanced kidney function (eGFR < 0.30 mu/min/1.73 m^2^) have been consistently excluded, the exact population with unequivocally evident CAN [57], with CKD being the most important factor predisposing to CAN [38,58]. This likely explains how, paradoxically, CKD has not been found to predict CAN in Hua’s study [8], in sharp contrast with the consensus regarding this predisposition, likely reflecting the loss of renal functional reserve with declining renal functional mass [53,59]. In the same fashion, patients with additional major factors predisposing to AKI following coronary interventions, such as hemodynamic instability [58], have also been excluded, further reducing the likelihood to include clinically significant CAN and the need for dialysis in these studies. With that in mind, and with the background noise of non-relevant confounders mentioned above potentially promoting increasing eGFR, the clinical significance of improved kidney function reported in these studies might be misleading.

In addition to the technical and methodological confounders mentioned above, the unexpected contradicting renal outcomes in patients subjected to CMs during cardiac interventions while on SGLT2is may also reflect diverse counteracting physiologic actions of SGLT2is regarding kidney function and parenchymal integrity. These effects, related to the interaction of diabetes, CM, SGLT2i and cardiac performance, detailed below, are schematically illustrated in Figure 1.

**Myocardial salvage and the preservation of the cardiorenal axis** might be an additional factor favoring patients treated with SGLT2is regarding renal outcome following coronary interventions, surpassing the potential direct harmful impact of SGLT2is on renal parenchymal integrity. Preserving cardiac function improves renal perfusion, reducing pre-renal components of renal dysfunction, while the prevention of backward failure reduces renal venous pressure, an important component of cardiorenal syndrome [40]. SGLT2i-related myocardial protection involves their osmotic diuretic and natriuretic effects, which lead to favorable cardiac hemodynamics, evidenced by reducing preload and afterload [60,61]. Furthermore, SGLT2is augment the diuretic effect of loop diuretics without causing electrolytes accompanied by the activation of the renin–angiotensin–aldosterone axis and the sympathetic nervous system. Additionally, SGLT2is may improve myocardial energetics, shifting myocardial (and renal) fuel metabolism away from energy-inefficient fat and glucose oxidation, toward the energy-efficient use of ketones as super fuels, hence improving myocardial (and renal) work efficiency and function [62]. Moreover, their ability to increase hematocrit assumedly increases oxygen delivery to the failing myocardium, while their anti-oxidative and anti-inflammatory action may attenuate cardiac damage and myocardial remodeling [60,61].

**Hypoxia tolerance** could be an additional explanation for the unexpected reduced risk of CAN in patients treated with SGLT2is. The repeated sublethal reduction of outer medullary pO_2_, invoked by the enhancement of distal tubular transport and oxygen consumption, could lead to the stabilization of the hypoxia-inducible factor (HIF) signal, a central regulator of adaptive and tissue-protective gene expression in response to hypoxia [63]. Regional hypoxia inhibits specific prolyl hydroxylases responsible for the initial step in the degradation of HIF-α sub-unit, leading to its cytoplasmic accumulation. This enables the binding of HIF-α and β sub-units, with subsequent nuclear translocation of the heterodimer and its attachment to DNA hypoxia response elements (HREs), generating hypoxia-adaptive responses. The up-regulation of HIF-mediated protective genes is considered a principal factor in the phenomenon of local, and likely remote, ischemia/hypoxia preconditioning [64]. Indeed, the renal HIF signal intensifies in experimental diabetes [23] and possibly confers tubular protection during acute hypoxic insults, such as the administration of CMs. This hypoxia tolerance mechanism may be further intensified by an additional decline of medullary pO_2_, caused by SGLT2is. Indeed, the attenuation of renal tubular hypoxic injury has been documented by hypoxia mimetic agents that block HIF prolyl hydroxylases, both ex vitro [65] and in vivo [20], and such compounds are already in use in the clinical practice, for instance in triggering HIF-dependent erythropoietin production [66].

In that respect, interestingly, analysis of the EMPA-REG OUTCOME trial showed that **increased hemoglobin** had the best association with renal protection in patients with cardiovascular disease treated with SGLT2is [67]. Another hint for activating the potentially protective HIF signal was the recently demonstrated renal preservation of another HIF-mediated gene, epidermal growth factor (EGF), in patients on SGLT2is, which was found to correlate with a slower progression of diabetic kidney disease [68].

There is ample evidence that SGLT2is may also **directly promote tissue protection** through a host of cellular adaptive responses [69,70]. As shown in cardiomyocytes, SGLT2is inhibit cardiac sodium transporters and alter ion homeostasis, probably via direct action on cardiac NHE and Late-INa flux, thus reducing Na+ and Ca2+ overload-mediated myocardial damage. They may also reduce inflammation and oxidative stress by inhibiting the differentiation of monocytes to macrophages and through promoting the polarization of macrophages from a pro-inflammatory M1 phenotype to an anti-inflammatory M2 phenotype. SGLT2is may also suppress the activation of inflammasomes and major pro-inflammatory factors. Finally, SGLT2is possibly improve renal cellular bioenergetics, as discussed above regarding myocardial salvage. Noteworthy, these tissue-protective properties are most apparent when cells are subjected to physiologic stress, including reactive oxygen species, inflammation, acidosis, hypoxia, high-saturated fatty acids, hypertension, hyperglycemia and sympathetic over stimulation. This likely takes place via modulating intracellular sodium homeostasis secondary to their inhibitory action on membranal Na^+^ loaders (NHE-1, SGLT and Nav1.5) [28,48]. It is, therefore, tempting to assume that **SGLT2is may also be renoprotective during exposure to CMs by directly affecting gene expression in tubular cells**, as recently shown in vitro in cultured tubular cell lines [71,72].

One may argue that since SGLT2is improve cortical oxygenation, the **attenuation of proximal tubular damage** during radiocontrast-enhanced imaging and interventions may improve renal outcome. There is some evidence that proximal tubular injury does also take place in CAN, with a rise in urine and plasma kidney ischemia molecule (KIM)-1, a proximal tubule-specific biomarker (though inconsistently or to a lesser sensitivity and specificity compared with NGAL) [73,74]. Proximal tubular cells indeed show vacuolar changes following the exposure to CMs. Yet, these changes are in fact invaginations of basolateral membranes, are transient and are not accompanied by other features of cell injury, and, most importantly, they consistently appear following the administration of CMs with no or minimal association whatsoever with kidney functional changes [18,32].

The direct tubular toxicity of CMs, in part by the enhanced formation of oxygen free radicals, may also exist in the pathogenesis of CAN, with a consequent inflammatory response [31,37,75]. SGLT2is increase diuresis, reducing the intraluminal concentration of CMs and shortening the urinary transit time along the nephron. This should reduce a possible tubular uptake of the contrast material and **diminish the likelihood of inflammation and direct tubular toxicity** [75]. Furthermore, SGLT2i-related diuresis conceivably **attenuates increased urine viscosity**, and reduces the accumulation of detached cells, leading to the lessening CM-induced rise in renal parenchymal hydrostatic pressure and improving the vasa recta blood flow and medullary oxygen supply [76].

## 6. Summary and Conclusions

Clinical studies suggest that the use of SGLT2is may reduce the risk of CAN, and this might be attributed to a host of renal protective mechanisms. Yet, with the possible confounders discussed above, we remain concerned regarding the potential dual impact of SGLT2is and CMs on the intensification of medullary hypoxia. This might be especially important in coronary interventions, where relatively large volumes of contrast material are used. Unequivocally, immediate cardiac interventions for STEMI should not be postponed due to SGLT2i pre-treatment. But what about elective procedures? Should we also delay such procedures for 24 h in high-risk patients with non-STEMI? Should we withhold the resumption of SGLT2is in these patients for a day or so following cardiac procedures?

We believe that, as of now, conclusions regarding the renal safety and salvage conferred by SGLT2is during radiocontrast studies should be restricted to selected patients, excluding those specifically predisposed to CAN, especially patients with advanced CKD, effective volume depletion or hemodynamic instability. For the time being, we call for caution regarding extending the liberal use of SGLT2is around and immediately following cardiac interventions to these patients at high risk and advocate for withholding SGLT2is in patients who are acutely ill and at high risk during radiocontrast studies and interventions, whenever possible. Similar warnings are likely relevant for patients who are high risk undergoing contrast-enhanced computerized tomography, even though smaller volumes of a radiocontrast medium are usually needed.

A recent article that has just been published indeed justify our call for caution [77]. Propensity score analysis in a cohort of diabetic patients undergoing coronary interventions revealed that the risk of CAN was markedly increased with the use of SGLT2is. Most interesting, the likelihood of CAN has been specifically restricted to short-term users of SGLT2is. This observation is in line with our concept of the risk of hypoxic medullary injury by the combination of CM and SGLT2is, counterbalanced by renoprotective mechanisms, activated by long-term usage of SGLT2is.

Further prospective controlled studies are needed to evaluate this dispute, particularly in patients who are high risk, and to reveal whether SGLT2is indeed provide renal protection or are hazardous during contrast-enhanced studies. These studies should be controlled for all clinical parameters believed to predispose to CAN, including hemodynamic status during and following the cardiac interventions [41,58], for parameters of glycemic control and for comorbidities and medications that affect renal oxygenation and function, such as diuretics or non-steroidal anti-inflammatory agents. Such prospective evaluations should also be extended to other populations, with the extending use of SGLT2is in patients without diabetes with heart failure or chronic kidney disease. This might also help elucidate a potential confounding impact of SGLT2is on glycemic control. Though we anticipate a class effect for all types of SGLT2is, outcomes should be evaluated separately for the various agents in use. A concomitant assessment of biomarkers of renal parenchymal injury is suggested in order to overcome the impact of confounders that may affect GFR without causing tubular damage. Furthermore, such studies should specifically be extended to patients most susceptible to CAN, as the hazardous or advantageous impact of such a combination is expected to be highly clinically relevant.

Until this dispute is settled, we propose to hold treatment with SGLT2is for 24 h before and after elective coronary interventions in patients with advanced CKD, and to postpone SGLT2i resumption for 24 h following urgent coronary studies and procedures in patients at high risk (principally advanced CKD and hemodynamic instability) or in patients administered with large volumes of CMs.

## Figures and Tables

**Figure 1 ijms-25-10759-f001:**
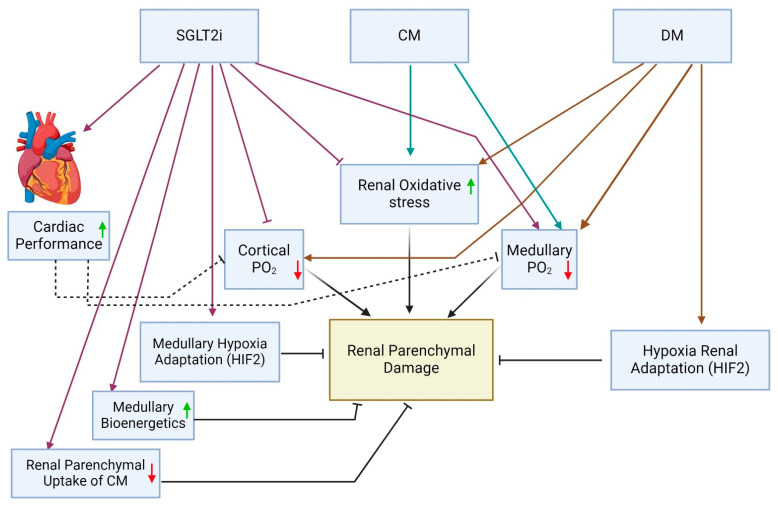
Suggested integrated impact of diabetes mellitus (DM), iodinated contrast medium (CM) and SGLT2 inhibitors (SGLT2is) on renal parenchymal oxygenation, oxidative stress and injury and a plausible SGLT2i-mediated renal protection through myocardial salvage. HIF—hypoxia-inducible factor 2; PO2—partial oxygen pressure.

**Table 1 ijms-25-10759-t001:** Characteristics of available studies exploring the impact of SGLT2 inhibitors (SGLT2is) on contrast-associated nephropathy among patients undergoing coronary interventions.

Study	No. of Patients	Study Population	Contrast Volume (mL)	Baseline eGFR	CAN Definition	Statistical Methods	AKI Outcomes in Patients Given SGLT2i
Kültürsay et al. [10]	130 of 295 on SGLT2is empagliflozin or dapagliflozin exposure for at least 6 months	Retrospective single-center study. Patients undergoing PCI for STEMI	Mean 291 and 265 mL in SGLT2i and control groups, respectively	≥30 mL/min/1.73 m^2^	A rise in sCr of ≥0.3 mg/dL above baseline within 48 h of contrast media exposure or an increase of at least 1.5 times the baseline value within 7 days	Doubly robust inverse probability weighted logistic regression analysis	AKI: lower occurrence, OR 0.86; 95% CI [0.76–0.98]; (*p* = 0.028)
Paolisso et al. [9]	111 of 646 on SGLT2i chronic SGLT2-I therapy (started at least 3 months before hospitalization)	Retrospective single-center study. Patients with AMI (STEMI and non-STEMI)	Median 180 mL [IQR 140–240]	≥30 mL/min/1.73 m^2^	Not clearly defined	Multivariable Cox regression model	AKI: lower occurrence, 5.4% vs. 13.1% (*p* = 0.022)
Özkan and Gürdoğan [12]	104 of 312 on SGLT2is, duration and timing of SGLT2 inhibitors were not indicated	Retrospective single-center study. Patients undergoing angiography with or without PCI	Mean 170 mL	≥30 mL/min/1.73 m^2^	sCr rise > 0.5 mg/dL or >25% above baseline within 48 h, or >1.5 times above baseline within 7 days or a urinary output of less than 0.5 mL/kg/h for at least 6 h	Multivariate binary logistic regression analysis	AKI: lower occurrence, OR 0.41, 95% CI [0.142–0.966]; *p* = 0.004
Hua et al. [8]	242 on SGLT2is matched with 242 non-users, canagliflozin, empagliflozin or dapagliflozin for at least 6 months till the date of PCI	Retrospective single-center study. Patients undergoing angiography with or without PCI	Mean 141 and 149 mL in the SGLT2i and control groups, respectively	serum creatinine ≤ 2.5 ng/dL	sCr rise ≥ 0.5 mg/dL or >1.25 times above baseline within 72 h	Propensity score matching followed by McNemar’s test	AKI: lower occurrence, OR 0.37; 95% CI [0.19–0.67]; (*p* < 0.01)
Meregildo-Rodriguez et al. [14]	512 of 2572 on SGLT2is, the mean time of SGLT2 I therapy duration was 7.3 ± 3 months	Meta analysis of 4 observational studies following coronary angiography with or without PCI	Not reported	Not reported	Absolute increase in sCr by 0.3 to 0.5 mg/dL or 25 to 50% relative increase within 48–72 h following coronary intervention	Meta analysis	AKI: lower occurrence, RR 0.37; 95% CI [0.24–0.58]
Feitosa MPM et al. [11]	22 patients on iSGLT-2 and 20 controls. The SGLT2i empagliflozin 25 mg daily was initiated at least 15 days before PCI and maintained until the end of the follow-up period	Prospective single-center open-label, randomized study of patients undergoing elective PCI	144 ± 66 mL in the SGLT2i users vs. 176 ± 54 mL in non-users	62.1 ± 22.5 mL/min in SGLT2i users and 68.2 ± 17.7 mL/min in non-users	A 25% increase in baseline creatinine or an absolute increase of 0.5 mg/dL between 48 and 72 h after contrast administration	Chi-square test	There was no difference in the primary endpoint of the study
Çabuk and Hazır [48]	133 patients with DM on SGLT-2i matched with 212 non-users, patients who were using an SGLT2 inhibitor (empagliflozin or dapagliflozin) for at least 6 months	Cross-sectional and single-center study. Patients underwent CAG and/or PCI	160.42 (±70.31) mL in the SGLT2i users vs. 158.72 (±81.24) mL in non-users	71.44 mL/min (57.04–86.22) vs. 66.08 mL/min (52.23–84.16)	Increase in serum creatinine of ≥0.5 mg/dL or an absolute increase of ≥25% from baseline 72 h after CM exposure	Wilcoxon signed-rank test	CA-AKI incidence was significantly lower in patients using SGLT2 inhibitors (9.0%) compared with non-users (26.4%, *p* < 0.001).

AMI—acute myocardial infarction; CAG—elective coronary angiography, CI—confidence interval; OR—odds ratio; PCI—primary percutaneous coronary intervention; STEMI—ST-elevation myocardial infarction; RR—relative risk; sCr—serum creatinine.

## Data Availability

No new data were generated or analyzed in support of this research.

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
