# Peer review of "SGLT2 Inhibitors and the Risk of Contrast-Associated Nephropathy Following Angiographic Intervention: Contradictory Concepts and Clinical Outcomes"

_ijms, 2024, doi:10.3390/ijms251910759_

Round 1

Reviewer 1 Report

Comments and Suggestions for Authors

SGLT2 inhibitors have gone beyond their initial indication of diabetes and are now used extensively to treat congestive heart failure and chronic kidney disease in patients without diabetes.  The authors have focused on acute kidney failure in the setting of post contrast radiologic procedures. Therre are many papers suggesting that the use of SGLT2 inhibitors is beneficial in reducing the incidence of acute kidney failure following these procedures.  The authors have produced a thoughtful analysis of the physiologic relationships created by SGLT2 inhibitor used and reposed the question of benefit versus harm.   They point out to numerous issues of patient selection to question the conventional wisdom of continuing or even initiating SGLT2 treatment in patients who may enter these procedures.  Their emphasis of characteristics of the study cohorts is well placed.  The concern here is that further elaboration of patient populations is necessary, particularly in light of the growing use of SGLT2 inhibitors in patients who do not have diabetes. The authors must rephrase their arguments paying particular attention to the absence of prospective studies in congestive heart failure and chronic renal disease patients.  In particular, they must deal with the potential effect of hyperglycemia on the outcomes.  We currently have an amalgam of patient cohorts.  Separating this out may be impossible at this stage, but they can make the recommendation.

Author Response

We thank reviewer 1 for his/her valuable comments.

Please find bellow itemized response to these comments:

Comments and Suggestions for Authors

SGLT2 inhibitors have gone beyond their initial indication of diabetes and are now used extensively to treat congestive heart failure and chronic kidney disease in patients without diabetes.  The authors have focused on acute kidney failure in the setting of post contrast radiologic procedures. There are many papers suggesting that the use of SGLT2 inhibitors is beneficial in reducing the incidence of acute kidney failure following these procedures.  The authors have produced a thoughtful analysis of the physiologic relationships created by SGLT2 inhibitor used and reposed the question of benefit versus harm.   They point out to numerous issues of patient selection to question the conventional wisdom of continuing or even initiating SGLT2 treatment in patients who may enter these procedures.  Their emphasis of characteristics of the study cohorts is well placed.  The concern here is that further elaboration of patient populations is necessary, particularly in light of the growing use of SGLT2 inhibitors in patients who do not have diabetes. The authors must rephrase their arguments paying particular attention to the absence of prospective studies in congestive heart failure and chronic renal disease patients.  In particular, they must deal with the potential effect of hyperglycemia on the outcomes.  We currently have an amalgam of patient cohorts.  Separating this out may be impossible at this stage, but they can make the recommendation.

Thank you for this comment. In the revised version, we now emphasize this point, calling for prospective randomized studies in diabetic patients, as well as in non-diabetics with CHF or CKD.

Reviewer 2 Report

Comments and Suggestions for Authors

- How does the duration and timing of SGLT2 inhibitor medication before contrast exposure affect the risk of contrast-associated nephropathy in patients with varying stages of CKD?
- Are there any genetic variants that influence the renal response to SGLT2 inhibitors in contrast-associated nephropathy, and how could these be factored into patient-specific risk assessments?

- Which specific biochemical pathways might SGLT2 inhibitors utilize to either exacerbate or mitigate medullary hypoxia-induced kidney damage in diabetic kidneys compared to non-diabetic kidneys?
- Regarding the cumulative risk of contrast-associated nephropathy (CAN), how do SGLT2 inhibitors interact with other nephrotoxic medications commonly used during angiography procedures, such as diuretics and antiplatelet agents?

- What is the relationship between the risk of CAN and the differing effects of specific SGLT2 inhibitors (e.g., canagliflozin vs. empagliflozin) on renal oxygenation and perfusion?
- How might chronic adaptive cellular responses induced by SGLT2 inhibitors, such as hypoxia-inducible factor (HIF) stabilization, protect against acute kidney injury when exposed to contrast agents?
- What connection exists between the vulnerability to CAN and the prevalence of microvascular complications, such as diabetic retinopathy, in patients using SGLT2 inhibitors?

Comments on the Quality of English Language

Minor editing is necessary.

Author Response

We thank reviewer 2 for his/her valuable comments.

Please find bellow itemized response to these comments:

Comment 1-How does the duration and timing of SGLT2 inhibitor medication before contrast exposure affect the risk of contrast-associated nephropathy in patients with varying stages of CKD?

Response: We have updated Table 1 according to your comment. Unfortunately, not all relevant studies referred to duration and timing of administered SGLT2 inhibitors in relation to PCI. We personally contacted responding authors and partially managed to obtain some data as incorporated in table 1.

Comment 2- Are there any genetic variants that influence the renal response to SGLT2 inhibitors in contrast-associated nephropathy, and how could these be factored into patient-specific risk assessments?

Response: We are unaware of genetic variants that affect the renal response to SGLT2 inhibitors. If you refer to congenital mutations that affect SGLT2 expression, this is likely to be diagnosed in the young age and the likelihood that such persons were included in the studies is conceivably negligible.

Comment 3- Which specific biochemical pathways might SGLT2 inhibitors utilize to either exacerbate or mitigate medullary hypoxia-induced kidney damage in diabetic kidneys compared to non-diabetic kidneys?

Response: In the revised submission, we have added figure 1, which summarizes the interactions between CM, SGLT2 inhibitors and DM, detailed in the text. We hope that this illustration may help clarifying the complex potential actions of SGLT2i in patients subjected to contrast materials. Furthermore, as now emphasized without getting into many details, exacerbation of medullary hypoxia is believed to by upstream to most biochemical pathways involved in CAN

Comment 4- Regarding the cumulative risk of contrast-associated nephropathy (CAN), how do SGLT2 inhibitors interact with other nephrotoxic medications commonly used during angiography procedures, such as diuretics and antiplatelet agents?

Response: This is indeed a thoughtful remark. Thank you. People freely use “nephrotoxicity” regarding CAN. Nevertheless, as outlined in the text, altered balance of renal microcirculation and oxygen consumption, leading to regional medullary hypoxia, is upstream to most pathways involved in the pathogenesis of CAN (Wang, Toxics 2024 Aug 22;12(8):620). Furosemide dramatically improves medullary oxygenation, and prevents medullary hypoxic injury in an animal model of CAN, as it inhibits medullary oxygen consumption, but with insufficient fluid replacement this can lead to pre-renal failure (without tubular damage) (Heyman, Am J Kidney Dis. 1989 Nov;14(5):377-85). Its protective impact in the prevention of CAN in high-risk individuals undergoing PCI may be appreciated in clinical trials using fusid as a part of protocols of forced diuresis, such as the RenalGuard studies (Brigouri, Circulation. 2011 Sep 13;124(11):1260-9). Unlike other NSAIDs (which intensify medullary hypoxia) low-dose aspirin is considered safe, whereas tPA enhances medullary blood flow (Heyman, Br J Pharmacol. 2004 Mar;141(6):971-8). Nevertheless, we are unaware of data regarding the combined effects of SGLT2 inhibitors with these medications or other agents used in PCI on renal microcirculation, oxygenation or tubular injury during contrast studies. Nevertheless, we now address your remark, underscoring a potential impact of other medications given during PCI.

Comment 5- What is the relationship between the risk of CAN and the differing effects of specific SGLT2 inhibitors (e.g., canagliflozin vs. empagliflozin) on renal oxygenation and perfusion?

 Response: An excellent comment, but regretfully, to our knowledge, there is no data comparing different SGLT2 inhibitors regarding their effect on renal microcirculation and oxygenation. We now mention this point in the call for prospective future studies.

Comment 6- How might chronic adaptive cellular responses induced by SGLT2 inhibitors, such as hypoxia-inducible factor (HIF) stabilization, protect against acute kidney injury when exposed to contrast agents?

Response: Hypoxia adaptation has been shown to attenuate acute hypoxic organ injury, and HIF stabilizers were shown to attenuate medullary hypoxia in isolated perfused kidneys (as quoted in the text, Rosenberger, NDT 2008). As medullary hypoxia plays a central role in CAN, conceivably hypoxia tolerance at the outer medulla, invoked by chronic administration of SGLT2 inhibitors may confer tissue protection at this region. This point is now better clarified in the text dealing with HIF and by figure 1 

Comment 7- What connection exists between the vulnerability to CAN and the prevalence of microvascular complications, such as diabetic retinopathy, in patients using SGLT2 inhibitors?

Response: We have previously outlined complex mechanisms by which diabetes and its complications predispose to CAN (quoted reference Heyman, Biomed Res Int 2013). It is not only microvascular structural changes, but also, altered microvascular flow, transport activity, tissue hypoxia, oxygen free radicals, altered NO availability, interstitial fibrosis, etc. There might be numerous possible interactions of such mechanisms and SGLT2 inhibitors, most of them beneficial (excluding the intensification of medullary hypoxia) but we are reluctant to extend on that, as this may be too speculative and out of focus, in an already extended discussion.   

Round 2

Reviewer 1 Report

Comments and Suggestions for Authors

The authors have issued a flag regarding potential for acute renal deterioration in certain classes of patients on SGLT2 inhibitor therapy undergoing the physiologic stress of angiographic procedures.   Their arguments are sound. At present there is no data one way or the other, but they offer a useful discourse.

Reviewer 2 Report

Comments and Suggestions for Authors

The paper can be accepted in its present form.